# Molecule Generation For Target Protein Binding with Structural Motifs

**Zaixi Zhang**[1,2]**, Yaosen Min**[3]**, Shuxin Zheng**[4]**, Qi Liu**[1,2∗]
1: Anhui Province Key Lab of Big Data Analysis and Application,
School of Computer Science and Technology, University of Science and Technology of China
2:State Key Laboratory of Cognitive Intelligence, Hefei, Anhui, China
3:Institute of Interdisciplinary Information Sciences, Tsinghua University, 4: Microsoft Research
zaixi@mail.ustc.edu.cn, minys18@mails.tsinghua.edu.cn
shuz@microsoft.com, qiliuql@ustc.edu.cn

## Abstract

Designing ligand molecules that bind to specific protein binding sites is a fundamental problem in structure-based drug design. Although deep generative models and geometric deep learning have made great progress in drug design, existing works either sample in the 2D graph space or fail to generate valid molecules with realistic substructures. To tackle these problems, we propose a **F**ragment-based **L**ig**A**nd **G**eneration framework (FLAG), to generate 3D molecules with valid and realistic substructures fragment-by-fragment. In FLAG, a motif vocabulary is constructed by extracting common molecular fragments (*i.e.,* motif) in the dataset. At each generation step, a 3D graph neural network is first employed to encode the intermediate context information. Then, our model selects the focal motif, predicts the next motif type, and attaches the new motif. The bond lengths/angles can be quickly and accurately determined by cheminformatics tools. Finally, the molecular geometry is further adjusted according to the predicted rotation angle and the structure refinement. Our model not only achieves competitive performances on conventional metrics such as binding affinity, QED, and SA, but also outperforms baselines by a large margin in generating molecules with realistic substructures. Our code is publicly available at https://github.com/zaixizhang/FLAG.

## 1 Introduction

Recent years have witnessed the great success of deep learning in drug design. Among the progress, deep generative models that aim to generate molecules with desirable physicochemical and pharmacological properties are of particular importance. These models range from string-based (Gómez-Bombarelli et al., 2018) and graph-based methods (Jin et al., 2018; Xie et al., 2021) to recent 3D geometry-based methods (Gebauer et al., 2019; Luo & Ji, 2021).

Molecule drugs can only affect certain biological functions and pathways by binding to the target proteins. However, the complexity of the context information, geometric constraints, and molecule-protein interactions bring great challenges. Therefore, few deep learning models have been developed to generate molecules that bind to specific protein binding sites (*a.k.a.* structure-based drug design). Early attempts modify the pocket-free models by incorporating scoring functions like docking scores between generated molecules and pockets to guide the ligand generation (Li et al., 2021). Another line of works convert the 3D pocket structures to molecular string or graph representations for conditional generation (Skalic et al., 2019; Xu et al., 2021a). They fail to model how molecules interact with their target proteins explicitly in 3D space. Recently, a series of 3D generative models are proposed to generate 3D molecules that bind to given protein pockets (Luo & Ji, 2021; Liu et al., 2022; Peng et al., 2022). They use 3D graph neural networks for context encoding and achieve equivariance. However, most of these works do not consider chemical priors and may generate

---

∗Qi Liu is the corresponding author.

invalid molecules with unrealistic substructures. Their atom-wise generation scheme also leads to inefficient molecule sampling.

In this work, we propose a novel **F**ragment-based **L**ig**A**nd **G**eneration framework (FLAG) for structure-based drug design, where the molecules are generated fragment-by-fragment. To generate 3D molecules, we first preprocess the dataset and extract molecular fragments with high occurrence frequencies (*i.e.,* motif) as "building blocks" for new molecules. At each generation step, a 3D graph neural network is first employed to encode the intermediate context information including the protein pocket and the intermediate molecular graph. Secondly, our model selects the focal motif, predicts the next motif type, and attaches the new motif to the generated molecule. Attaching motifs in 3D space is a great challenge. Inspired by the fact that the flexibility of molecular geometries lies largely in the degree of rotatable bond (A bond in a molecule is rotatable if cutting this bond creates two connected components of the molecule, each of which has at least two atoms) (Axelrod & Gomez-Bombarelli, 2022), we employ cheminformatics tools (Bento et al., 2020) to efficiently determine the bond lengths/angles and trains neural networks to predict torsion angles. Leveraging this insight can significantly reduce the searching space of atom/motif coordinates. For example, the CrossDocked dataset (Francoeur et al., 2020) has, on average, $m = 24$ heavy atoms, corresponding to a $3m$-dimensional euclidean space, but only around 5 torsion angles of rotatable bonds. Furthermore, the rotation angle is predicted to further adjust the geometries of generated molecules. Inspired by force fields in physics (Rappé et al., 1992), a novel structure refinement is finally applied to optimize the molecule structures.

We conduct extensive evaluations to evaluate our approach. Experimental results show that: (1) our method is able to generate diverse drug-like molecules with high binding affinity to target proteins; (2) FLAG is much faster than most of the baseline methods at sampling new molecules; (3) thanks to the design of fragment-based generation, our method outperforms baselines by a large margin on generating valid molecules with realistic substructures.

## 2 RELATED WORK

**Motif-based Molecule Generation.** To generate more valid molecules with realistic substructures, many models adopt prior knowledge of chemical motifs, also known as fragments or rationales, as building blocks to generate or optimize molecules (Jin et al., 2018; 2020a; Podda et al., 2020; Jin et al., 2020b; Chen et al., 2021a; Seo et al., 2021; Xie et al., 2021; Chen et al., 2021b; Guo et al., 2022; Flam-Shepherd et al., 2022). For example, JT-VAE (Jin et al., 2018) first decomposes the molecular graphs into junction trees, where each node in the tree represents a substructure of the molecule. Then JT-VAE adopts the variational autoencoder as the framework and learns to reconstruct the molecular graph fragment-by-fragment. Similarly, RationaleRL extracts rationales that lead to different properties of molecules by MCTS. Then it is trained to expand rationales to complete molecular graphs with reinforcement learning. However, the aforementioned methods cannot generate 3D molecules directly and consider the complicated context information of binding sites.

**3D Molecule Generation.** With the development of geometric deep learning, many recent works explore 3D molecular geometry generations with given 2D molecular graphs (Mansimov et al., 2019; Simm & Hernandez-Lobato, 2020; Luo et al., 2021b; Shi et al., 2021; Ganea et al., 2021; Xu et al., 2021b; 2022), or from scratch (Gebauer et al., 2019; Hoogeboom et al., 2022; Nesterov et al., 2020; Gebauer et al., 2022; Luo & Ji, 2021; Satorras et al., 2021). Comparatively, the task of structure-based drug design is more challenging. Firstly, the 2D molecular graph is unknown. Secondly, the generated molecules should fit well with the binding pockets with high binding affinity. Finally, the aforementioned works usually deal with small organic molecules and may be insufficient to generate 3D drug-like molecules with larger molecule weights. For more detailed discussions on molecule generation, we recommend readers refer to the comprehensive survey (Du et al., 2022).

**Structure-based Drug Design.** Structure-based drug design aims to generate 3D molecules that bind to specific binding sites. LiGAN (Ragoza et al., 2022) first approaches this problem using a conditional variational autoencoder trained on atomic density grid representations of protein-ligand structures. Then the molecular structures of ligands are constructed by atom fitting and bond inference from the generated atom densities. As a preliminary work, LiGAN employs 3D CNN as the encoder, which does not satisfy the desirable equivariance property. The follow-up works achieve

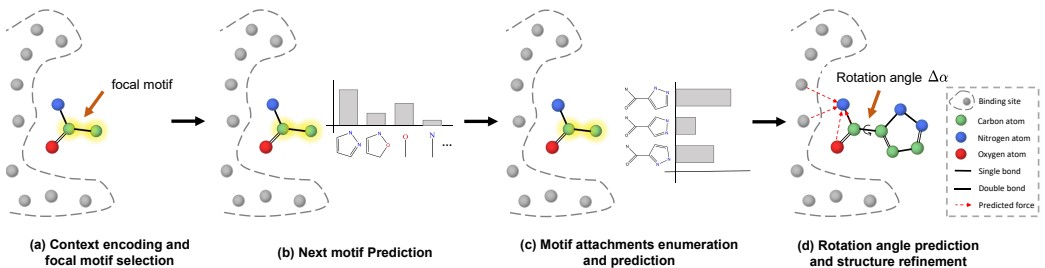

Figure 1: An illustration of one generation step of FLAG. There are mainly four parts whose details are shown in Sec.3.

equivariance by leveraging graph neural networks to encode the context information (Luo & Ji, 2021; Liu et al., 2022; Peng et al., 2022). For example, (Luo & Ji, 2021) uses SchNet (Schütt et al., 2017) to encode the 3D context of binding sites and estimate the probability density of atom's occurrences in 3D space. The atoms are sampled auto-regressively until there is no room for new atoms. GraphBP (Liu et al., 2022) adopts the framework of normalizing flow (Rezende & Mohamed, 2015) and constructs local coordinate systems to predict atom types and relative positions. Pocket2Mol (Peng et al., 2022) adopts the geometric vector perceptrons (Jing et al., 2021) and the vector-based neural network (Deng et al., 2021) as the context encoder. It also explicitly considers the influence of chemical bonds. However, most of the above methods do not consider the prior knowledge of chemical motifs and may generate molecules with unrealistic or distorted substructures.

We note that some recent works (Green et al., 2021; Powers et al., 2022) are similar to our model that adopts fragment-based methods for ligand generation/optimization. (Green et al., 2021) optimizes the binding affinity by predicting fragments to add. (Powers et al., 2022) expands a small molecule fragment into a larger drug-like molecule binding to a given protein pocket. Compared with these methods, FLAG does not require the information of starting fragments and can automatically construct the motif vocabulary. Moreover, the novel structure refinement module enables FLAG to adjust the generated molecules flexibly.

## 3 METHOD

### 3.1 OVERVIEW

Our goal is to generate valid 3D molecular structures that can fit and bind to specific protein binding site. The 3D geometry of a molecule (*i.e.,* ligand) can be represented as $\mathcal{G} = \{(\boldsymbol{a}_i, \boldsymbol{r}_i)\}_{i=0}^n$. Similarly, a binding site of protein can be defined as a set of atoms $\mathcal{P} = \{(\boldsymbol{b}_j, \boldsymbol{s}_j)\}_{j=1}^m$. Here, we use $n$ and $m$ to represent the numbers of atoms in the molecule and in the binding site, respectively. In $\mathcal{G}$ and $\mathcal{P}$, $\boldsymbol{a}_i$ and $\boldsymbol{b}_j$ are one-hot vectors denoting the atom types and $\boldsymbol{r}_i, \boldsymbol{s}_j \in \mathbb{R}^3$ indicate the 3D cartesian coordinates. Formally, our objective is to learn a conditional generative model $p(\mathcal{G}|\mathcal{P})$ to capture the conditional distribution of protein-ligand pairs.

In our work, we formulate the generation of molecules in the given binding pocket as a sequential generation process. Let our generation model be $\phi$ and the intermediate generated molecule at the $t$-th step be $\mathcal{G}^{(t)}$, the generation process can be summarized as below:

$$\mathcal{G}^{(t)} = \phi(\mathcal{G}^{(t-1)}, \mathcal{P}), \ t > 1 \tag{1}$$

$$\mathcal{G}^{(1)} = \phi(\mathcal{P}), \ t = 1. \tag{2}$$

Different from previous works, we generate molecules motif-by-motif, *i.e.,* a set of atoms from the new motif are included into $\mathcal{G}^{(t)}$ instead of a single atom. Specifically, there are mainly four parts in one generation step, including (a) context encoding and focal motif selection, (b) next motif prediction, (c) motif attachments enumeration and selection, and (d) rotation angle prediction and structure refinement (Figure.1). The details are demonstrated in the rest part of this section.

In this section, we first introduce the motif extraction procedure in Sec. 3.2. In Sec. 3.3 and Sec. 3.4, we will describe the architecture of the encoder and the 3D molecule generation process respectively. In Sec. 3.5, we derive the training objective introduces the training scheme in detail.

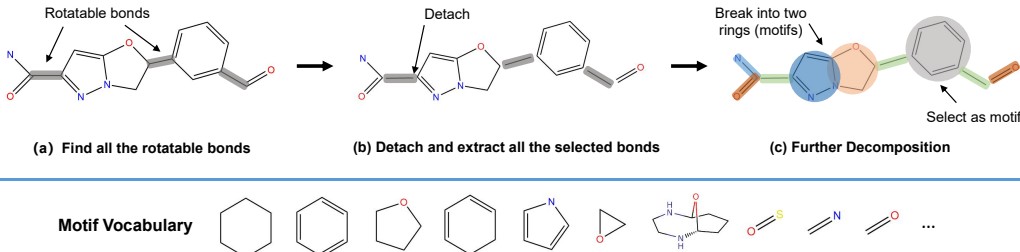

Figure 2: Illustration of Motif Extraction. More details are shown in Sec.3.2.

## 3.2 MOTIF EXTRACTION

To decompose molecules and extract motifs, a molecule can also be represented as a three-dimensional graph $\mathcal{G} = (\mathcal{V}, \mathcal{R}, \mathcal{E})$ with $\mathcal{V}$ as atoms set, $\mathcal{R}$ as atom coordinates set, $\mathcal{E}$ as covalent bonds set. A motif $\mathcal{M}_i = (\mathcal{V}_i, \mathcal{R}_i, \mathcal{E}_i)$ is defined as a subgraph of molecule $\mathcal{G}$. Given a molecule, we extract its motifs $\mathcal{M}_1, \cdots, \mathcal{M}_n$ such that their union covers the entire molecular graph: $\mathcal{V} = \bigcup_i \mathcal{V}_i$, $\mathcal{R} = \bigcup_i \mathcal{R}_i$, and $\mathcal{E} = \bigcup_i \mathcal{E}_i$. The motif extraction mainly contains the following steps:

- Firstly, extract and detach all the rotatable bonds that will not violate the chemical validity.
- Then molecule $\mathcal{G}$ is then broken into disconnected fragments $\mathcal{G}_1, \cdots, \mathcal{G}_N$.
- We select $\mathcal{G}_i$ as motif if its occurrence in the whole training set is more than $\tau$. If $\mathcal{G}_i$ is not selected as a motif, we further decompose it into finer rings and bonds and select them as motifs in $\mathcal{G}$.

After preprocessing the whole training dataset, we obtain a vocabulary of motifs $V_{\mathcal{M}}$. The motif extraction and vocabulary construction procedure are illustrated in Figure.2. In the generation process, the geometry of motifs (*i.e.,* bond lengths and angles) can be determined by cheminformatics tools such as RDkit (Bento et al., 2020).

## 3.3 ENCODER

To generate molecules conditioned on the binding pocket, it is important to capture the context information with the context encoder. In our work, a context 3D graph $\mathcal{C}^{(t-1)} = \mathcal{G}^{(t-1)} \bigcup \mathcal{P}$ is firstly constructed by connecting atoms within certain cutoff distances. A 3D graph neural network is employed to encode $\mathcal{C}^{(t-1)}$. The first layer of the encoder is a linear layer which maps atomic attributes to initial embeddings $\boldsymbol{h}_k^{(0)}$. Then, we have $L$ feature aggregation layers in our 3DGNN. The aggregation for each atom $k$ at the $l$-th layer ($1 \leq l \leq L$) can be formulated as:

$$\boldsymbol{h}_k^{(t,l)} = \boldsymbol{h}_k^{(t,l-1)} + \sum_{u \in \mathcal{N}(k)} \boldsymbol{h}_u^{(t,l-1)} \odot \mathrm{MLP}^l(\boldsymbol{e}_{\mathrm{RBF}}(d_{uk})), \tag{3}$$

where $\mathcal{N}(k)$ denotes the neighbors of the $k$-th atom in $\mathcal{C}^{(t-1)}$, $\mathrm{MLP}^l(\cdot)$ is a multi-layer perceptron, and $\odot$ denotes the element-wise multiplication. The embedding of pairwise distance $d_{uk}$ is obtained with radial basis functions ($\boldsymbol{e}_{\mathrm{RBF}}(\cdot)$), such as Gaussian Functions (Schlichtkrull et al., 2018) and spherical Bessel functions (Gasteiger et al., 2019). Since our encoder is based on the atom attributes and pairwise distances, it is rotationally and translationally invariant. Note that more advanced 3DGNNs such as DimeNet (Gasteiger et al., 2019) and SphereNet (Liu et al., 2021) can be employed as encoders in future works. We do not use them in the current version of FLAG, considering the computational efficiency and GPU memory budget.

## 3.4 3D MOLECULE GENERATION

The following procedures are applied to generate the new motif in each generation step.

**Focal Motif Selection**: To predict the next motif, we need to first select a focal motif to which the next motif attaches with. We employ two auxiliary atom-wise classifiers: protein atom classifier (for

$t = 1$) and molecule atom classifier (for $t \geq 2$) for the selection. (1) At the first step ($t = 1$), all the known context information is the binding pocket. The protein atom classifier takes the hidden representations of protein atoms as input, and predicts whether new ligand atoms can be generated within 4 Å. (2) For $t \geq 2$, the molecule atom classifier selects a focal atom from the ligand atoms generated in the previous $t - 1$ steps. The motif that the focal atom belongs to is chosen as the focal motif. If no atom/motif is selected as focal, the molecule generation is completed.

Overall, the classifiers take the representations of atoms as input and utilize two MLPs to predict the selection probabilities. We describe how to train these two auxiliary classifiers in Sec.3.5.

**Next Motif Prediction**: Given the focal motif $\mathcal{M}_f$, the label of the next motif is predicted as:

$$q = \text{Softmax}(\text{MLP}^{\mathcal{M}}(\text{Emb}(\mathcal{M}_f), \sum_{i \in \mathcal{M}_f} h_i)) \tag{4}$$

where $q$ is the distribution over the motif vocabulary $V_{\mathcal{M}}$, $\text{Emb}(\mathcal{M}_f)$ denotes the embedding of the focal motif, and $\sum_{i \in \mathcal{S}_f} h_i$ is the sum of the atom embeddings in the focal motif. Since there is no focal motif at the first step ($t = 1$), we regard *no motif* as a special motif type and also learn its embedding in training.

**Motif Attachments Enumeration and Prediction:** With the predicted motif, the next step is to attach the new motif to the generated molecule. Note that this step is not deterministic since there are potentially many attachment configurations (See Figure.1). Our goal here is to select the most appropriate attachment. Specifically, we enumerate different *valid* attachments and form a candidate set $C$. We employ GIN (Xu et al., 2019) as the scoring function $f_a$ over the candidate molecular graphs and train it to select the most appropriate molecule attachment:

$$\mathcal{G}^{(t)} = \arg\max_{\mathcal{G}' \in C} f_a(\mathcal{G}'). \tag{5}$$

Note that by our design, any two motifs share at most two atoms, so we only need to merge at most two atoms or one bond in the process of motif attachment. By pruning chemically invalid molecules and merging isomorphic graphs, we have $|C| \approx 3$ on the CrossDocked dataset. Therefore, the attachment and scoring will not be a burden on the computational efficiency.

After selecting the molecule attachment graph, we also need to assemble the new motif to the intermediate molecule in the 3D space. The assembling process of the first motif is challenging since there is no reference motifs to attach with and the relative position to the binding pocket is unknown. A straightforward method is to randomly place the first motif near the focal protein atom. However, such a strategy is noisy and predicted coordinates can be far from the optimal solutions. Following (Jin et al., 2022), we use a distance-based initialization strategy, which is more accurate and stable than random initialization. Specifically, a distance matrix $\boldsymbol{D} \in \mathbb{R}^{(n'+m') \times (n'+m')}$ is set as:

$$\boldsymbol{D}_{i,j} = \begin{cases} \|\boldsymbol{s}_i - \boldsymbol{s}_j\| & i, j \leq n' \\ \text{MLP}^d(\boldsymbol{h}_i^{(0)}, \boldsymbol{h}_j^{(0)}) & i \leq n', j > n' \\ \|\boldsymbol{r}_i - \boldsymbol{r}_j\| & i, j > n', \end{cases} \tag{6}$$

where $n'$ and $m'$ denote the number of sampled protein atoms for reference and the number of atoms in the first molecular motif. The distances between the protein atoms and motif atoms can be directly calculated. For the distances between molecular and protein atoms, we use $\text{MLP}^d$ for prediction with the pairwise atom attributes as the input. With the distance matrix $\boldsymbol{D}$, we can obtain the coordinates of atoms by eigenvalue decomposition of its Gram matrix (Crippen & Havel, 1978):

$$\tilde{\boldsymbol{D}}_{i,j} = 0.5(\boldsymbol{D}_{i,1}^2 + \boldsymbol{D}_{1,j}^2 - \boldsymbol{D}_{i,j}^2), \quad \tilde{\boldsymbol{D}} = \boldsymbol{U}\boldsymbol{S}\boldsymbol{U}^{\top} \tag{7}$$

where $\boldsymbol{S}$ is a diagonal matrix with eigenvalues in descending order. The coordinate of each atom $\boldsymbol{r}_i$ is calculated as:

$$\tilde{\boldsymbol{r}}_i = [\boldsymbol{X}_{i,1}, \boldsymbol{X}_{i,2}, \boldsymbol{X}_{i,3}], \quad \boldsymbol{X} = \boldsymbol{U}\sqrt{\boldsymbol{S}}. \tag{8}$$

Note that even though predicted coordinates $\{\tilde{\boldsymbol{r}}_i\}$ retain the original distance $\boldsymbol{D}$, they are located in a different coordinate system. Therefore, we apply the Kabsch algorithm (Kabsch, 1976) to find a rigid body transformation $\{\boldsymbol{R}, \boldsymbol{t}\}$ that aligns the predicted protein coordinates $\{\tilde{\boldsymbol{r}}_1, \cdots, \tilde{\boldsymbol{r}}_{n'}\}$ with the reference coordinates $\{\boldsymbol{r}_1, \cdots, \boldsymbol{r}_{n'}\}$. Lastly, the coordinates of the first motif are calculated as:

$$\boldsymbol{r}_i = \boldsymbol{R}\tilde{\boldsymbol{r}}_i + \boldsymbol{t}, \; i > n'. \tag{9}$$

For generation steps with $t > 1$, the coordinates of the attached motifs are determined and aligned similarly with RDkit (Bento et al., 2020) and Kabsch algorithm (Kabsch, 1976). The atoms in the focal motif are used as the reference atoms. If the focal motif contains a rotatable bond, the new motif has an additional degree of freedom. The torsion angle should be further predicted and adjusted.

**Rotation Angle Prediciton**: After attaching the new motif and obtaining the initial coordinates, we apply the encoder again to get the updated hidden representations. Let $X, Y$ denote the two end atoms of the rotatable bond (specifically, let $Y$ denote the atom connecting the new motif). The change of the torsion angle, $\Delta\alpha$ is predicted as:

$$\Delta\alpha = \mathrm{MLP}^\alpha(h_X, h_Y, h_\mathcal{G})\mathrm{mod}2\pi, \tag{10}$$

where $h_X$ and $h_Y$ indicate the embeddings of $X$ and $Y$; $h_\mathcal{G}$ denotes the embedding of the molecule, which is obtained with a sum pooling. Since the predicted angle is based on the representations from the equivariant encoder, $\Delta\alpha$ is also rotationally and translationally invariant. Finally, the coordinates of the atoms in the new motif are updated by rotating $\Delta\alpha$ around line $XY$. The implementation details are included in Appendix A.

**Structure Refinement:** According to the design of our generation scheme, at each step $t > 1$, we fix the coordinates of $\mathcal{G}^{(t-1)}$, predict and attach the new motif. However, we find the generated co-ordinates may be inaccurate and lead to sub-optimal generated molecules for the following reasons: (1) the distance-based initialization at $t = 1$ may be inaccurate because the pairwise distances are predicted independently without considering higher-order interaction terms; (2) at $t > 1$, we fix the coordinates of $\mathcal{G}^{(t-1)}$ and ignore the interactions between $\mathcal{G}^{(t-1)}$ and the new motif. Therefore, we propose to incorporate an structure refinement process at each step of generation. The refinement procedure should be equivariant as the coordinates of $\mathcal{G}^{(t-1)}$ and $\mathcal{P}$ can be rotated and translated arbitrarily. Inspired by force fields in physics (Rappé et al., 1992), we calculate the *force* between atoms. Specifically, after placing the new motif at each step $t$, we additionally use our encoder to compute the atom embeddings $h_i^{(t)}$. The *force* between atoms in $\mathcal{G}^{(t)}$ is calculated as:

$$\boldsymbol{g}_{i,j}^{(t)} = g(h_i^{(t)}, h_j^{(t)}, \boldsymbol{e}_{\mathrm{RBF}}(\|\boldsymbol{r}_i - \boldsymbol{r}_j\|)) \cdot \frac{\boldsymbol{r}_i - \boldsymbol{r}_j}{\|\boldsymbol{r}_i - \boldsymbol{r}_j\|}, \tag{11}$$

where $\boldsymbol{e}_{\mathrm{RBF}}(\|\boldsymbol{r}_i - \boldsymbol{r}_j\|)$ is the radial basis distance encoding, the function $g$ is a feed-forward neural network with a scalar output, and $\frac{\boldsymbol{r}_i - \boldsymbol{r}_j}{\|\boldsymbol{r}_i - \boldsymbol{r}_j\|}$ is the normalized verctor from the $j$-th atom to the $i$-th atom. For the The *force* between $\mathcal{G}^{(t)}$ and $\mathcal{P}$, we only calculate the alpha carbons $C_\alpha$ in $\mathcal{P}$ for computational efficiency. Similarly, we have:

$$\boldsymbol{g}_{i,k}^{(t)} = g(h_i^{(t)}, h_k^{(t)}, \boldsymbol{e}_{\mathrm{RBF}}(\|\boldsymbol{r}_i - \boldsymbol{s}_k\|)) \cdot \frac{\boldsymbol{r}_i - \boldsymbol{s}_k}{\|\boldsymbol{r}_i - \boldsymbol{s}_k\|}. \tag{12}$$

Finally, the coordinates of $\mathcal{G}^{(t)}$ is updated as:

$$\boldsymbol{r}_i^{(t)} \leftarrow \boldsymbol{r}_i^{(t)} + \frac{1}{|\mathcal{G}^{(t)}|} \sum_{j \neq i} \boldsymbol{g}_{i,j}^{(t)} + \frac{1}{|\mathcal{P}_{C_\alpha}|} \sum_k \boldsymbol{g}_{i,k}^{(t)}, \tag{13}$$

where $|\mathcal{G}^{(t)}|$ denotes the number of molecular atoms at step $t$; $|\mathcal{P}_{C_\alpha}|$ denotes the number of $C_\alpha$ atoms in the binding pocket. In the Appendix B, we perform ablation studies to show the effectiveness of structure refinement.

## 3.5 TRAINING

In the training stage, we first extract motifs for molecules with the method described in Sec.3.2. we randomly mask the motifs of molecules and train the model to recover the masked ones. Specifically, for each pocket-ligand pair, we sample a mask ratio from the uniform distribution $U[0, 1]$ and mask the corresponding number of structural motifs. The motifs are generated in a breadth-first order and the root motif is set as the motif closest to the pocket. The atoms that have valence bonds to the masked motifs are defined as focal atom candidates. If all molecular atoms are masked, the focal atoms are defined as protein atoms that have masked ligand atoms within 4 Å.

For the motif type prediction, we use cross-entropy loss for the classification, denoted as $\mathcal{L}_{motif}$. For the focal atom/motif prediction, we use a binary cross entropy loss $\mathcal{L}_{focal}$ for the classification of focal atoms. For the distance prediction and structure refinement, we minimize an MSE loss $\mathcal{L}_d$ with respect to the pairwise distances. We add Gaussian noise to the ground truth coordinates in the training stage and refine the structure with the predicted force. For the motif attachment, we maximize the log-likelihood of predicting the correct molecular graph:

$$\mathcal{L}_{attach} = -f_a(\mathcal{G}_t) + \log \sum_{\mathcal{G}' \in C/\mathcal{G}_t} \exp(f_a(\mathcal{G}')), \quad (14)$$

where $\mathcal{G}_t$ denotes the ground truth and $C$ is the set of possible candidates. As for the torsion angle prediction, we fit angles with von Mises distributions with $\mathcal{L}_\alpha$ similar to (Senior et al., 2020). In the training process, we aim to minimize the sum of the above loss functions:

$$\mathcal{L} = \mathcal{L}_{motif} + \mathcal{L}_{focal} + \mathcal{L}_{attach} + \mathcal{L}_d + \mathcal{L}_\alpha. \quad (15)$$

## 4 EXPERIMENTS

### 4.1 EXPERIMENTAL SETTINGS

**Dataset:** Following (Luo et al., 2021a) and (Peng et al., 2022), we use the CrossDocked dataset (Francoeur et al., 2020) which contains 22.5 million protein-molecule structures. We filter out data points whose binding pose RMSD is greater than 1 Å and molecules that can not be sanitized with RDkit (Bento et al., 2020), leading to a refined subset with around 160k data points. We use mmseqs2 (Steinegger & Söding, 2017) to cluster data at 30% sequence identity, and randomly draw 100,000 protein-ligand pairs for training and 100 proteins from remaining clusters for testing. For evaluation, we randomly sample 100 molecules for each protein pocket in the test set.

**Baselines:** We compare FLAG with four state-of-the-art baseline methods: one method based on 3D CNNs (LiGAN) (Ragoza et al., 2022) and another three methods based on 3D GNNs (AR, GraphBP, and Pocket2Mol) (Luo et al., 2021a; Liu et al., 2022; Peng et al., 2022).

**Model:** The number of layers $L$ in context encoder is 6, and the hidden dimension is 256. The model is trained with the Adam optimizer at a learning rate of 0.0001. The batch size is 4 and the number of total training iterations is 1,000,000. The standard deviation of the added Gaussian noise to the ligand coordinates is 0.2. The cutoff distance in the context encoder is set to 10 Å. The threshold $\tau$ in motif extraction is set to 100 in the default setting.

**Metrics:** We choose metrics that are widely used in previous works (Luo et al., 2021a; Peng et al., 2022; Liu et al., 2022) to evaluate the qualities of the sampled molecules: (1) **Vina Score** measures the binding affinity between the generated molecules and the protein pockets; (2) **High Affinity** is calculated as the percentage of pockets whose generated molecules have higher affinity to the references in the test set; (3) **QED** measures how likely a molecule is a potential drug candidate; (4) **Synthesizability (SA)** represents the difficulty of drug synthesis (the score is normalized between 0 and 1 and higher values indicate easier synthesis); (5) **LogP** is the octanol-water partition coefficient (LogP values should be between -0.4 and 5.6 to be good drug candidates (Ghose et al., 1999)); (6) **Lipinski (Lip.)** calculates how many rules the molecule obeys the Lipinski's rule of five (Lipinski et al., 2012); (7) **Sim. Train** represents the Tanimoto similarity (Bajusz et al., 2015) with the most similar molecules in the training set; (8) **Diversity (Div.)** measures the diversity of generated molecules for a binding pocket (It is calculated as 1 - average pairwise Tanimoto similarities). (9) **Time** records the time cost to generate 100 valid molecules for a pocket. In our work, the Vina Score is calculated by QVina (Trott & Olson, 2010; Alhossary et al., 2015) and the chemical properties are calculated by RDKit (Bento et al., 2020) over the valid molecules. Before feeding to Vina, the generated molecular structures are firstly refined by universal force fields (Rappé et al., 1992).

### 4.2 RESULTS

In Table. 1, we show the mean values along with standard deviations of the above metrics. Generally, our method achieves competitive or better performance compared with the baseline methods. Even though our method is not explicitly optimized for the binding affinity, FLAG succeeds at generating molecules with higher affinity than reference molecules over 58% of the binding site, which is quite

Table 1: Comparing the molecular properties of the test set and the generated molecules by different methods. The best results are bolded.

| Methods | Vina Score (kcal/mol, ↓) | High Affinity(↑) | QED (↑) | SA (↑) | LogP | Lip. (↑) | Sim. Train (↓) | Div. (↑) | Time (↓) |
|---|---|---|---|---|---|---|---|---|---|
| Testset | -7.180 ± 2.27 | - | 0.483 ±0.23 | 0.710 ±0.15 | 0.932 ±2.71 | 4.373 ±1.17 | - | - | - |
| LiGAN | -6.129 ±1.63 | 0.214 ±0.12 | 0.395 ±0.24 | 0.612 ±0.18 | -0.126 ±2.58 | 4.013 ±1.25 | 0.455 ±0.24 | 0.620 ±0.13 | - |
| AR | -6.231 ±1.66 | 0.236 ±0.13 | 0.489 ± 0.21 | 0.676 ± 0.17 | 0.261 ±2.33 | 4.757 ±0.42 | 0.417 ± 0.21 | 0.675 ± 0.16 | 18962 ±9794 |
| GraphBP | -7.012 ±1.75 | 0.430 ±0.25 | 0.471 ±0.18 | 0.706 ±0.25 | 0.439 ±2.05 | 4.776 ±0.45 | 0.410 ±0.23 | 0.651 ±0.14 | 1059.4 ±536.7 |
| Pocket2 Mol | -7.113 ±2.42 | 0.536 ±0.26 | **0.522** ±0.23 | 0.733 ±0.22 | 1.215 ±2.39 | 4.896 ±0.24 | 0.385 ±0.20 | 0.693 ±0.18 | 2655.1 ±1740 |
| Ours | **-7.247** ±2.25 | **0.580** ±0.24 | 0.495 ±0.17 | **0.745** ±0.16 | 0.630 ±2.38 | **4.943** ±0.14 | **0.370** ±0.22 | **0.704** ±0.15 | **1047.6** ±681.5 |

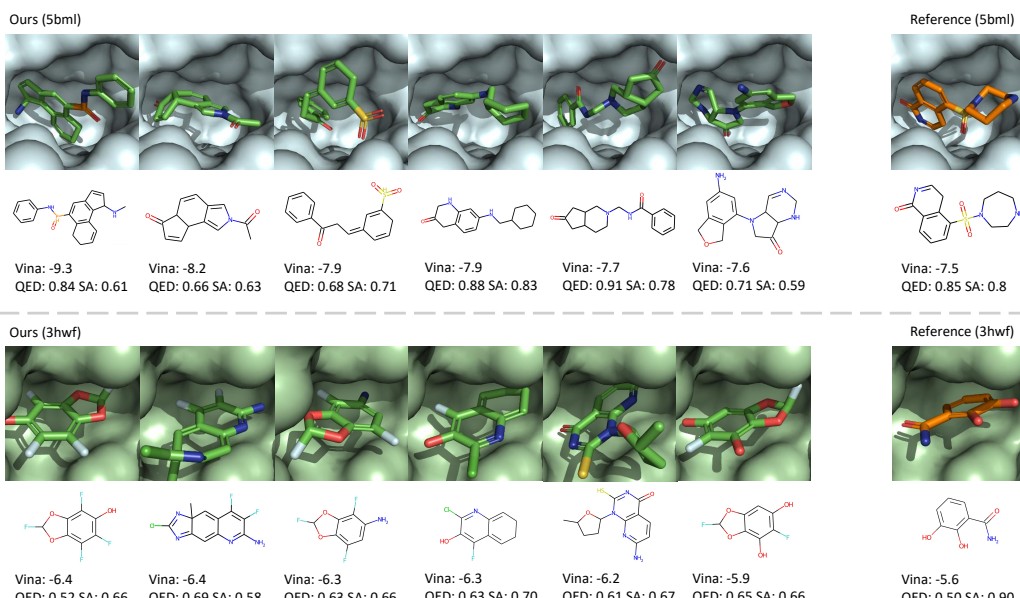

Figure 3: Examples of generated 3D molecules that have higher binding affinity than reference molecules. Lower Vina score indicates higher binding affinity.

impressive. When it comes to drug potentials (QED, SA, LogP, and Lipinski), we observe that FLAG obtains the best result on SA and Lipinski, which indicates that molecules generated by FLAG are more likely to be drug candidates. Moreover, our model manages to generate diverse molecules with the lowest similarities to the molecules in the training dataset. This shows that our model can generalize to new protein pockets and does not just memorize the training dataset. Table.1 also shows the generation time comparisons among 3D GNN-based methods. Our model has comparable sampling efficiency with GraphBP, and is much faster than AR and Pocket2Mol. This is because our method generates molecules fragment-by-fragment and has much fewer generation steps. Due to page limits, more discussions of molecule generations are shown in Appendix B.

In Figure 3, we provide two case studies and show several examples of generated 3D molecules that have a higher binding affinity (lower vina scores) than their corresponding reference molecules. Firstly, we observe that our generated molecules with higher affinity are largely different from reference molecules in structure. This further validates that our model can generate diverse and novel molecules to bind target proteins, instead of just imitating or modifying reference molecules. Moreover, the QED and SA scores are comparable or even higher than the reference molecules.

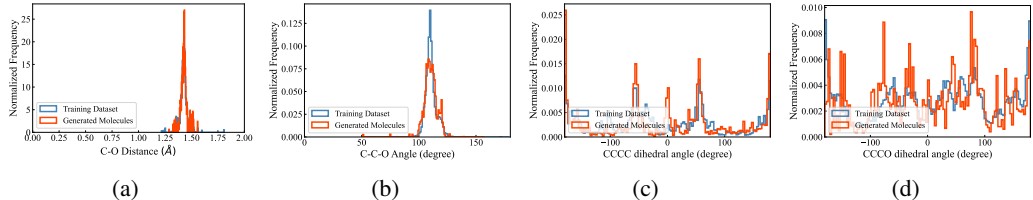

Figure 4: Distributions for C-O bond lengths (a), CCO bond angles (b), CCCC dihedral angles (c), and CCCO dihedral angles (d) in the training set and generated molecules.

Table 2: The KL divergence of the bond angles (upper part) and dihedral angles (lower part) between the test set and the generated molecules. The lower letters represent the atoms in the aromatic rings.

| Angles | LiGAN | AR | GraphBP | Pocket2 Mol | FLAG w/o SR | FLAG w/o T | FLAG |
|--------|-------|------|---------|-------------|-------------|------------|------|
| CCC    | 7.26  | 2.17 | 1.67    | 0.88        | 0.30        | 0.25       | **0.24** |
| CCO    | 8.31  | 1.95 | 2.02    | 0.93        | 0.17        | 0.19       | **0.12** |
| CNC    | 6.90  | 2.36 | 1.26    | 0.55        | 0.24        | 0.22       | **0.18** |
| NCC    | 7.53  | 3.59 | 2.10    | 0.68        | 0.29        | 0.26       | **0.21** |
| CC=O   | 6.85  | 3.56 | 1.23    | 0.80        | 0.23        | 0.27       | **0.15** |
| CCCC   | 1.26  | 0.81 | 0.79    | **0.73**    | 0.80        | 0.82       | 0.77 |
| cccc   | 6.43  | 8.26 | 5.41    | 4.30        | 0.33        | 0.31       | **0.30** |
| CCCO   | 2.34  | 1.55 | 1.09    | 1.14        | 0.57        | 0.93       | **0.51** |
| Cccc   | 4.98  | 8.27 | 3.13    | 2.35        | 0.58        | 0.51       | **0.48** |
| CC=CC  | 6.01  | 5.86 | 3.08    | 3.40        | 1.54        | 1.76       | **1.22** |

## 4.3 SUBSTRUCTURE ANALYSIS

Substructure analysis is needed to further evaluate the generated molecules Peng et al. (2022). In Table. 2, we measure the Kullback-Leibler (KL) divergence between the distributions of the bond angles/ dihedral angles in the test set and the generated molecules by different methods. The lower the values, the better the methods can capture the distributions of the dataset. In Table. 2, we can observe that our method has much lower KL divergences than the baseline methods over nearly all the angle types. This demonstrates that our method can keep the geometric attributes of the data well and generate molecules with realistic substructures. This again shows the advantage of the fragment-wise generation scheme. To further investigate how well our model learns the distribution of substructures, we show the bond length distributions (C-O single bond), bond angle distributions (CCO chains), and two sets of dihedral angle distributions (CCCC and CCCO chains) in Figure.4. We have the following observations: Firstly, the distributions of bond length/angle show clear clusters, indicating the values of bond length/angle are largely fixed. This also verifies the rationality to use chemical priors and tools to determine the bond lengths/angles. Secondly, the distributions of dihedral angles are more complex and it is difficult to pre-specify them when attaching new motifs. Therefore, we predict and adjust the torsion angle based on the context information in FLAG. Finally, the distributions of generated molecules by FLAG align well with the training set, indicating that the bond distances/angles and dihedral angles are accurately modeled and reproduced.

## 5 CONCLUSION

In this paper, we propose a Fragment-based LigAnd Generation framework, known as FLAG, to generate 3D molecules with valid and realistic substructures fragment-by-fragment. FLAG not only achieves competitive performances on conventional metrics such as binding affinity but also outperforms baselines by a large margin on generating realistic structures with speed-ups. Potential future works include: (1) exploring more powerful context encoders such as Graph Transformer Zhang et al. (2022b); (2) building more robust and interpretable SBDD models Zhang et al. (2021b;a; 2022d;a;c); (3) using pre-training to further improve performance Zhang et al. (2021d;c).

ACKNOWLEDGEMENTS

This research was partially supported by a grant from the National Natural Science Foundation of China (Grant No. 61922073).

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

# A IMPLEMENTATION DETAILS

## A.1 IMPLEMENTATION OF BASELINES

To implement the baselines including LiGAN [1], AR[2], GraphBP[3], and Pocket2Mol[4], we use the open-source codes and follow their default settings. The methods are trained on the same data split for a fair comparison. All the experiments are conducted on Ubuntu Linux with V100 GPUs. The codes are implemented in Python 3.8 and Pytorch 1.10.0. To sample molecules, we apply 5 independent runs with random seeds. Our code is publicly available at: `https://github.com/zaixizhang/FLAG`.

## A.2 DEFINITION OF BOND LENGTH AND BOND ANGLE

For a bond $AB$, its bond length is defined as $d(A, B) \overset{def}{=} \|\boldsymbol{r}_A - \boldsymbol{r}_B\|$, where $\boldsymbol{r}_A$ and $\boldsymbol{r}_B$ denote the 3D coordinates. For a bonded chain $ABC$, the bond angle is defined as $\cos\angle ABC \overset{def}{=} \frac{(\boldsymbol{r}_A - \boldsymbol{r}_B) \cdot (\boldsymbol{r}_C - \boldsymbol{r}_B)}{\|\boldsymbol{r}_A - \boldsymbol{r}_B\| \cdot \|\boldsymbol{r}_C - \boldsymbol{r}_B\|}$.

## A.3 DEFINITION OF DIHEDRAL ANGLE

The count-clockwise (CCW) dihedral angle between two intersecting half-planes $ABC$ and $ABD$ with an common edge $AB$ is calculated as [5]:

$$\angle(ABC, ABD) \overset{def}{=} atan2(\|\boldsymbol{b}_2\|\langle \boldsymbol{b}_1, \boldsymbol{b}_2 \times \boldsymbol{b}_3 \rangle, \langle \boldsymbol{b}_1 \times \boldsymbol{b}_2, \boldsymbol{b}_2 \times \boldsymbol{b}_3 \rangle), \tag{16}$$

where $\boldsymbol{b}_1 = \boldsymbol{r}_A - \boldsymbol{r}_C, \boldsymbol{b}_2 = \boldsymbol{r}_B - \boldsymbol{r}_A, \boldsymbol{b}_3 = \boldsymbol{r}_D - \boldsymbol{r}_B$.

## A.4 CANONICAL DEFINITION OF TORSION ANGLE

As introduced by (Ganea et al., 2021), the torsion angle can be defined in a canonical way. Let $X, Y$ be the coordinates of the two end atoms and $\{P_i\}, \{Q_j\}$ be the coordinates of their neighbors respectively. Let $\Delta_{ij} \overset{def}{=} \angle(XYP_i, XYQ_j)$ denote the counter-clockwise dihedral angle. $s_{ij} \overset{def}{=} [cos(\Delta_{ij}), sin(\Delta_{ij})]^T$. Let $c_{ij}$ be real coefficients such that $s \overset{def}{=} \sum_{i,j} c_{ij} s_{ij} \in \mathbb{R}^2$ is not a null vector. The torsion angle $\alpha$ is defined: $\alpha \overset{def}{=} atan2(\frac{s}{\|s\|})$.

In our method, instead of predicting the exact value of $\alpha$, we predict the change of the torsion angle $\Delta\alpha$ and update the coordinates by the rotation matrix described in Appendix A.6.

## A.5 CALCULATION OF KL DIVERGENCE

To calculate two distributions of bond/dihedral angles, we set one degree as a bin and calculate the normalized frequencies (180 bins for bond angles and 360 bins for dihedral angles). The KL divergence between two distributions $p(\theta)$ and $q(\theta)$ is calculated as:

$$KL = \sum_i p(\theta_i) \cdot log(\frac{p(\theta_i)}{q(\theta_i)}). \tag{17}$$

## A.6 DETAILS OF MOTIF ROTATION

In the generation of new motifs, if the focal motif is rotatable and the rotation angle $\Delta\alpha$ is known, we use the following rotation matrix $R_{3\times3}$ and the translation vector $t_{3\times1}$ to update the coordinates of the new motif. Let $X, Y$ denote the two end atoms of the rotatable bond ($Y$ denote the atom

---

[1] `https://github.com/mattragoza/LiGAN`
[2] `https://github.com/luost26/3D-Generative-SBDD`
[3] `https://github.com/divelab/GraphBP`
[4] `https://github.com/pengxingang/Pocket2Mol`
[5] `https://en.wikipedia.org/wiki/Dihedral_angle`

Table 3: The ratio of the molecules containing different rings in the test set and those generated by different methods. The results closest to the ratios of the test set are bolded.

| Ring Size | Test Set | LiGAN | AR | GraphBP | Pocket2Mol | Ours |
|---|---|---|---|---|---|---|
| 3 | 0.029 | 0.354 | 0.495 | 0.330 | 0.005 | **0.020** |
| 4 | 0.000 | 0.223 | 0.004 | 0.146 | **0.000** | **0.000** |
| 5 | 0.440 | 0.384 | 0.289 | 0.367 | 0.396 | **0.433** |
| 6 | 0.839 | 0.319 | 0.650 | 0.738 | 0.814 | **0.852** |
| 7 | 0.012 | 0.066 | 0.037 | 0.024 | 0.061 | **0.023** |
| 8 | 0.000 | 0.019 | 0.007 | 0.010 | 0.005 | **0.003** |
| 9 | 0.000 | 0.011 | 0.003 | 0.004 | 0.003 | **0.002** |

connecting the new motif) and $r_X$ and $r_Y$ be their coordinates. Let $n$ denotes the normalized directional vector $\frac{r_Y - r_X}{\|r_Y - r_X\|}$ and $n_x, n_y$ and $n_z$ be its components along the $x, y$ and $z$ axis. Let $x_0, y_0$, and $z_0$ be the three components of $r_X$. The rotation matrix and translation vector are:

$$R_{3\times3} = \begin{bmatrix} n_x^2 K + cos(\Delta\alpha) & n_x n_y K - n_z sin(\Delta\alpha) & n_x n_z K + n_y sin(\Delta\alpha) \\ n_x n_y K + n_z sin(\Delta\alpha) & n_y^2 K + cos(\Delta\alpha) & n_y n_z K - n_x sin(\Delta\alpha) \\ n_x n_z K - n_y sin(\Delta\alpha) & n_y n_x K + n_x sin(\Delta\alpha) & n_z^2 K + cos(\Delta\alpha) \end{bmatrix}, \quad (18)$$

$$t_{3\times1} = \begin{bmatrix} (x_0 - n_x M)K + (n_z y_0 - n_y z_0)sin(\Delta\alpha) \\ (y_0 - n_y M)K + (n_x z_0 - n_z x_0)sin(\Delta\alpha) \\ (z_0 - n_z M)K + (n_y x_0 - n_x y_0)sin(\Delta\alpha) \end{bmatrix}, \quad (19)$$

where $K = 1 - cos(\Delta\alpha)$ and $M = n_x x_0 + n_y y_0 + n_z z_0$. The coordinates $r_i$ in the motif are updated as:

$$r_i' = R r_i + t \quad (20)$$

### A.7 MORE DETAILS OF MOTIF ATTACHMENT

To determine the 3D coordinates of the new motif and attach it to the generated molecule, we leverage RDkit and Kabsch algorithm. The Kabsch algorithm (Kabsch, 1976) is a widely-used method to calculate the optimal rotation matrix that minimizes the RMSD (root mean squared deviation) between two paired sets of points. The atoms in the focal motif are selected as the reference atoms. We first use RDkit to get the 3D coordinates of the new motif and reference atoms. Since the coordinates of the new motif and the intermediate molecule $\mathcal{G}^{(t-1)}$ are located in different coordinate frames, we employ Kabsch algorithm to align them. Finally, we update the molecule coordinates by appending the coordinates of the new motif.

## B MORE EXPERIMENTAL RESULTS

### B.1 MORE ANALYSIS OF SUBSTRUCTURES

In Table.3, we show the ratios of the molecules containing different rings in the test set and the generated molecules by different methods. We can observe that the molecules generated by FLAG have the most similar ratios of rings of different sizes with the test set. This again shows the advantage of FLAG to learn and reproduce the distributions of substructures in the dataset.

### B.2 ABLATION STUDIES

In Table.4, we show the results of ablation studies regarding the structure refinement module (FLAG w/o SR) and the torsion angle module (FLAG w/o T). Specifically, in FLAG w/o SR, we remove the structure refinement in each generation step; in FLAG w/o T, we do not further rotate the motif for adjustment. In Table.4, we can observe that FLAG achieves the best performance on binding affinity and drug-likeness metrics (QED, SA, LogP, Lip.), verifying the effectiveness of the two modules.

Table 4: Ablation studies of the structure refinement module and the torsion angle module. FLAG w/o SR and FLAG w/o T denotes removing the structure refinement and torsion angle prediction and rotation procedure respectively.

| Methods | Vina Score (kcal/mol, ↓) | High Affinity(↑) | QED (↑) | SA (↑) | LogP | Lip. (↑) | Sim. Train (↓) | Div. (↑) | Time (↓) |
|---|---|---|---|---|---|---|---|---|---|
| FLAG | **-7.247** | **0.580** | **0.495** | **0.745** | 0.630 | **4.943** | **0.370** | **0.704** | 1047.6 |
| FLAG w/o SR | -7.016 | 0.437 | 0.465 | 0.710 | 0.692 | 4.925 | 0.396 | 0.651 | 1018.9 |
| FLAG w/o T | -6.945 | 0.356 | 0.486 | 0.706 | 0.733 | 4.927 | 0.407 | 0.689 | **1013.5** |

Moreover, with these two modules, FLAG can generate more unique and diverse molecules. Admittedly, structure refinement and torsion prediction bring more computational overhead. However, the overhead is acceptable considering the gain in generation performance.

## B.3 HYPER-PARAMETER ANALYSIS

Here, we mainly discuss the influence of hyper-parameter $\tau$ on the generation performance. In Figure.5, we vary $\tau$ from 0 to 500 and show the average Vina scores of the generated molecules by FLAG and its variants. Lower vina scores indicate higher binding affinity and better generation quality. We have the following observations. Firstly, our methods are robust to the choice of $\tau$. For example, FLAG has the largest vina score when $\tau = 0$. The value is only 0.2 larger than the lowest vina score at $\tau = 100$. Moreover, an appropriate value $\tau$ is important for FLAG to achieve the best performance. Lower $\tau$ typically indicates a larger size of motif vocabulary with redundant motifs. On the other hand, larger $\tau$ indicates a smaller size of motif vocabulary with finer extracted motifs (*e.g.,* single bonds), which may lead to too many generation

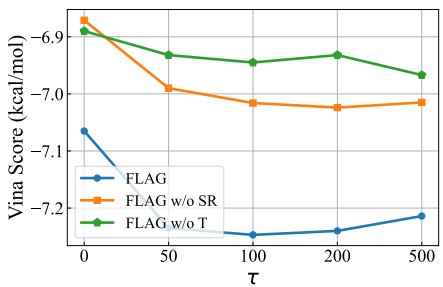

Figure 5: Hyper-parameter analysis with respect to $\tau$. Lower Vina scores indicate higher binding affinity and better generation quality.

steps and invalid generated molecules. Finally, an interesting observation is that FLAG w/o T has the best performance at $\tau = 500$. This is probably due to the reason that finer motifs offer more flexibility when the torsion angle module is removed in FLAG.

## B.4 GENERATED MOLECULES WITH LOWER AFFINITIES THAN THE REFERENCE

In Figure. 6, we show a pocket (3tym) where FLAG fails to generate molecules with higher affinities than the reference. FLAG fails to generate molecules with higher binding affinities probably due to the following reasons: (1) The geometry of this binding pocket (3tym) is more complex and some generated molecules fail to occupy the whole pocket like the reference molecule. (2) Some generated molecules accidentally collide with the pocket, which is unrealistic in nature.

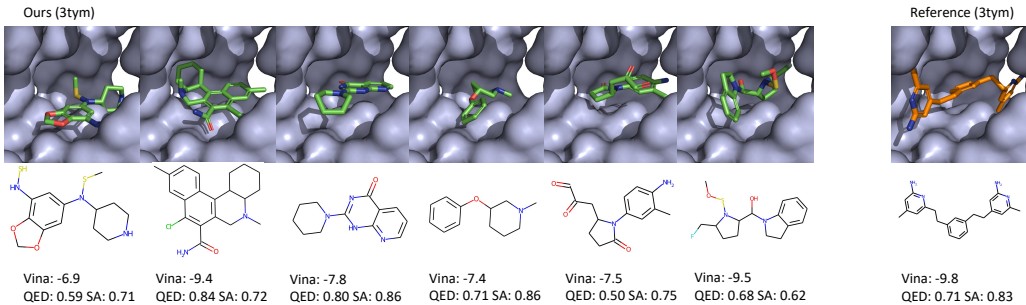

Figure 6: Examples of generated 3D molecules that have lower binding affinity than reference.

