# OpenReview forum: "Molecule Generation For Target Protein Binding with Structural Motifs"
_ICLR.cc/2023/Conference — ICLR 2023 poster_

### Official Review · Reviewer_nStG · 2022-10-23

**Confidence:** 4
**Correctness:** 3
**Technical Novelty And Significance:** 3
**Empirical Novelty And Significance:** Not applicable
**Recommendation:** 8

**Clarity, Quality, Novelty And Reproducibility:**

**Clarity**

The work is well-written and easy to follow. However, there are sections where I couldn't properly understand what has been done from the text alone. As for the writing style, some symbols are undefined, some occasional typos, and improper use of citations, but nothing of real concern. See below for details.

**Quality**

Overall, the work is medium-to-high quality. The modeling is fairly complicated, but I'd expect that since we are dealing with a very difficult task. The evaluation in particular is done with proper care (i.e. comparing to the baselines trained with the same data splits, which is not to be taken for granted in this field) and the results are convincing.

**Novelty**

The work reuses known concepts and modeling tools and puts them together in an innovative manner (at least, to my knowledge).

**Reproducibility**

The work is **not** reproducible. The code is not provided and the authors state that it will be released upon acceptance.

**Strength And Weaknesses:**

**Strengths**

- Impressive experimental results
- Substructure refinement appears innovative and its importance is confirmed through ablation studies

**Weaknesses**
- not reproducible


**Summary Of The Paper:**

The paper presents a model for generating molecules (represented as a 3D structure) that bind to a specific protein site (also represented in 3D). The authors propose a fragment-by-fragment sequential approach, which is summarized in the following steps:

- substructure (or motif) vocabulary creation
- context encoding (the context is initially just the binding site, then it's binding site + existing molecule)
- focal motif selection (i.e. selecting the initial motif / the motif used to expand the existing molecule)
- next motif prediction (i.e. selecting the motif to attach from the vocabulary
- motif attachment and orientation
- substructure refinement

In the experiments, the authors show that the model generates molecules with high binding affinity, sometimes even better than the reference, while also possessing high drug-likeness.

**Summary Of The Review:**

In general, I find this paper a solid contribution, probably worthy of ICLR. The most disappointing part is that I cannot look at the code, which to me makes it irreproducible.

Besides this, I have several questions I'd like the authors to answer, and some comments/suggestions/corrections.
Please acknowledge that I am not an expert in structure-based drug design (though I know the general problem and main approaches).

In no particular order:
- What is the vocabulary size? Do 2D-isomorphic motifs with different geometry get encoded as two different "words" in the vocabulary?
- You say that after pruning chemically invalid molecules and _merging isomorphic graphs_ you reduce the set of candidates attachments. How do you handle merging isomorphic graphs, considering that it can be computationally expensive in certain cases?
- What are the valid, novel, and unique fractions of the molecules generated for the test set?
- Can you show examples where the model fails to generate molecules that bind to a given target (as in Fig. 3)?
- Why did you use QVina to evaluate the Vina scores, instead of more recent alternatives (GNina)?
- You state that all the generated molecules were further refined with universal force fields before being fed to Vina. Did the molecules generated with the baselines get the same treatment?
- Why did you use GIN as a scoring function for candidate attachments?
- At times, the notation is confusing. For example, in Eq. 4), the $\mathrm{MLP}$ function has two arguments, while in Eq. 9 it has three. I thought that would mean concatenation, but then when in the equation where you define the distance matrix (which by the way is not numbered) the $\mathrm{MLP}^d$ uses $concat$ to describe a concatenated input. Please correct or explain.
- you say that $\boldsymbol{e}_{\mathrm{RBF}}$ can be either a Gaussian function or a spherical Bessel function. Which is the actual used in this work?
- you should use \citep instead of \cite for the citations.
- there is a missing closing parenthesis in Eq. 10

Finally, I think there are two citations that should be considered in Section 2 when you talk about motif-based generation.
- Podda et al. A Deep Generative Model for Fragment-Based Molecule Generation. AISTATS 2020
- Seo et al., Fragment-based molecular generative model with high generalization ability and synthetic accessibility. https://arxiv.org/abs/2111.12907

**Overall judgment**

I am giving a score of 6 for the moment being, will wait for the authors' replies to see whether there's space to increase my score.

**EDIT**

Since the authors kindly responded to all my questions and most of the questions raised by the other reviewers, I am changing my vote to an 8.

---

### Official Review · Reviewer_WWrH · 2022-10-23

**Confidence:** 4
**Correctness:** 3
**Technical Novelty And Significance:** 3
**Empirical Novelty And Significance:** 3
**Recommendation:** 6

**Clarity, Quality, Novelty And Reproducibility:**

The paper is well written and organized clearly. Therefore, the clarity and the quality of this paper is good. However, the authors do not illustrate key differences between this FLAG model and other fragment-based methods. Therefore, the novelty may be not strong enough. And the reproducibility is not very reliable and the contribution to the drug design community is limited since the authors do not provide code implementations of this model.

**Details Of Ethics Concerns:**

No.

**Strength And Weaknesses:**

Strength:
1.	It is an encouraging attempt since it is a quite difficult task to generate target-based molecules in 3D space.
2.	This method is of fragment-based molecular generation, which is in accord with functional groups in organic chemistry.
3.	Since the assembling process of the first motif and rotation angle prediction are subtly hard to deal with, this paper provides a tentative solution.


Weakness:
1.	The paper does not conduct the experiment or provide sufficient evidence about target specificity of the FLAG model, i.e. the comparison between the model with target information and the model without target information. It is possible that FLAG model just fits the structural distribution of the whole ligand library rather than the distribution conditioned on the specific target.
2.	The authors do not illustrate key differences between this FLAG model and other fragment-based methods, such as “Fragment-Based Ligand Generation Guided By Geometric Deep Learning On Protein-Ligand Structure” and “Scalable Fragment-Based 3D Molecular Design with Reinforcement Learning”. Besides, the authors say, “However, in these works, starting fragments are required and the motif vocabulary needs to be constructed by experts, limiting their practical applications.” However, introducing experts’ knowledge to generate drug-like molecules in order to avoid uncommon or toxic fragments is quite reasonable in drug design. This statement is invalidated.
3.	As for the motif extraction step, there are some doubts when it comes to find all the rotatable bonds. Rotatable bonds are usually just single bonds in organic chemistry. If rotatable bonds are under the condition that “where either u or v is part of a ring”, how does this model to deal with the situation that several single bonds are connected end-to-end?
4.	The “Motif Attachments Enumeration” way is quite similar to JT-VAE.
5.	It is not clearly demonstrated when the authors introduced the motivation of RBF in the “Encoder” part.
6.	In the “Structure Refinement”, it is not well illustrated that “the force between atoms” proposed by authors has true physical meaning or not.


**Summary Of The Paper:**

In order to design ligand molecules that bind to specific protein binding sites, the authors propose a Fragment-based LigAnd Generation framework (FLAG), to generate 3D molecules with valid and realistic substructures fragment-by-fragment. A motif vocabulary is constructed by extracting common molecular fragments (i.e., motif) in the dataset. At each generation step, a 3D graph neural network is first employed to encode the intermediate context information. Then, the model selects the focal motif, predicts the next motif type, and attaches the new motif. The bond lengths/angles can be quickly and accurately determined by cheminformatics tools. Finally, the molecular geometry is further adjusted according to the predicted rotation angle and the structure refinement. In their conducted experiments, results shows that their method is able to generate diverse drug-like molecules with good binding affinity to target proteins, is faster than most of the baseline methods at sampling new molecules and outperforms baselines by a large margin on generating valid molecules with realistic substructures.

**Summary Of The Review:**

In order to design ligand molecules that bind to specific protein binding sites, the authors propose a Fragment-based LigAnd Generation framework to generate 3D molecules with valid and realistic substructures fragment-by-fragment. Some design ways of 3D molecule generation are tentative and fairly inspiring. The paper is well written and organized clearly but the novelty may be not strong enough since the authors do not illustrate key differences between this FLAG model and other fragment-based methods. And the contribution to the drug design community is limited since the authors do not provide code implementations of this model.

After reading the responses, I raise my score a bit to 6.

---

> ### Author Response · Authors · 2022-11-12
> **Response to Reviewer WWrH (1/2)**
>
> We thank the reviewer for the valuable suggestions! As for the comments:
>
> Comment1: The paper does not conduct the experiment or provide sufficient evidence about target specificity of the FLAG model, i.e. the comparison between the model with target information and the model without target information. It is possible that FLAG model just fits the structural distribution of the whole ligand library rather than the distribution conditioned on the specific target.
>
> Response1: Thanks for the insightful question! In Table.1, we report the Sim. Train represents the Tanimoto similarity with the most similar molecules in the training set. Compared with baseline methods, FLAG has the least Sim. Train, which indicates that FLAG can generate molecules conditioned on the binding pocket instead of just reproducing the molecules in the training dataset.
> Following the reviewer’s suggestion, we additionally did experiments to generate molecules without target information. Specifically, we replace the context embedding with randomly initialized vectors and set the hyper-parameters as the default setting. Then we calculate the average Vina scores of the generated molecules to the pockets in the test set. The Vina score is only -5.314, much worse than -7.247 of FLAG with target information. This verifies the target specificity of FLAG.
>
>
> Comment2: The authors do not illustrate key differences between this FLAG model and other fragment-based methods, such as “Fragment-Based Ligand Generation Guided By Geometric Deep Learning On Protein-Ligand Structure” and “Scalable Fragment-Based 3D Molecular Design with Reinforcement Learning”. Besides, the authors say, “However, in these works, starting fragments are required and the motif vocabulary needs to be constructed by experts, limiting their practical applications.” However, introducing experts’ knowledge to generate drug-like molecules in order to avoid uncommon or toxic fragments is quite reasonable in drug design. This statement is invalidated.
>
> Response2: Thanks for the valuable suggestion! We will illustrate the differences between FLAG and previous works more clearly. Compared with “Fragment-Based Ligand Generation Guided By Geometric Deep Learning On Protein-Ligand Structure” , FLAG does not require the type, geometry, and location of the starting fragments. FLAG can also automatically construct the motif vocabulary and has the advantage of a novel structure refinement module. On the other hand, “Scalable Fragment-Based 3D Molecular Design with Reinforcement Learning” aims to generate 3D molecules and does not consider binding to protein pockets.
> We admit that introducing experts’ knowledge to generate drug-like molecules in order to avoid uncommon or toxic fragments is reasonable and our statement may be not accurate enough. We have modified our statement in the revised paper.
>
> Comment3: As for the motif extraction step, there are some doubts when it comes to find all the rotatable bonds. Rotatable bonds are usually just single bonds in organic chemistry. If rotatable bonds are under the condition that “where either u or v is part of a ring”, how does this model to deal with the situation that several single bonds are connected end-to-end?
>
> Response3: Thanks for your question! In the training and generation process, all single bonds that are not in rings are regarded as rotatable bonds. The description of the motif extraction step may not be accurate enough. The condition“where either u or v is part of a ring” is just used to firstly fragment the molecule and extract rings. We have modified the motif extraction step to make it clearer.

---

> ### Author Response · Authors · 2022-11-12
> **Response to Reviewer WWrH (2/2)**
>
> Comment4: The “Motif Attachments Enumeration” way is quite similar to JT-VAE
>
> Response4: The Motif Attachments Enumeration module in FLAG is different from that in JT-VAE. Specifically, in JT-VAE, the Motif Attachments Enumeration is used to reproduce molecular graphs that underlie the predicted junction tree. As for the scoring function, JT-VAE uses a message-passing method on the junction tree to extract information.
> On the contrary, our method FLAG uses Motif Attachments Enumeration to attach the intermediate molecular graph with the predicted motif. Different from JT-VAE, we employ a more powerful GNN, GIN as the scoring function. It scores the enumerations based on attached graphs instead of tree structures.
> We will modify our paper to make the differences clearer in the revised paper.
>
> Comment5: It is not clearly demonstrated when the authors introduced the motivation of RBF in the “Encoder” part.
>
> Response5: RBF is introduced to encode the pairwise distances between atoms [1].  3DGNN can better capture the geometric information with the RBF module. We will demonstrate the motivation of RBF more clearly in the revised paper.
>
> [1] Schütt et al. Schnet: A continuous-filter convolutional neural network for modeling quantum interactions. NeurIPS, 2017
>
> Comment6: In the “Structure Refinement”, it is not well illustrated that “the force between atoms” proposed by authors has true physical meaning or not.
>
> Response6: To optimize molecular geometries, traditional approaches mainly rely on force field refinement, where coordinates of atoms are sequentially updated based on the forces acting on each atom. Inspired by force fields in physics, we propose the structure refinement module to efficiently adjust the geometry of the generated molecules. The basic idea is to learn the pairwise interactions between atoms by neural networks, which can be viewed as pseudo-forces and has true physical meanings.
> We will illustrate the background and physical meanings of structure refinement more clearly in the future version.
>
> Comment7: the contribution to the drug design community is limited since the authors do not provide code implementations of this model.
>
> Response7: We have uploaded our code to https://anonymous.4open.science/r/FLAG-5C01/ for reference. It is our hope to benefit the drug design community. We are cleaning and organizing the code and will open-source the codes of FLAG with detailed documentation after acceptance.

---

### Official Review · Reviewer_xsvH · 2022-10-24

**Confidence:** 4
**Correctness:** 3
**Technical Novelty And Significance:** 3
**Empirical Novelty And Significance:** 2
**Recommendation:** 5

**Clarity, Quality, Novelty And Reproducibility:**

Clarity: Good. The paper is overall well-written and organized.

Quality: Okay.

Novelty: Medium. The proposed method is based on existing methods. The main contribution is to combine motif-based generation with target-based generation. The synergy of torsion angle prediction is interesting but not novel.

Reproducibility: N/A, the code has not been released.


**Strength And Weaknesses:**

Positives:  The author provides a clear and complete formulation of the problem and conducts various experiments to demonstrate the effectiveness of his design. As the experimental results of the paper suggest, fragment-based molecular generation does yield more rational structures.

Negatives: The technical contributions of this work are somewhat trivial. Motif vocabulary construction is a common strategy and has been well-explored in the field of drug design, and both predicting the torsion angle and selecting the focal motif are not proposed for the first time.  The interaction with the target was designed similarly to the previous method.

Questions:
1. Could the authors explain why the results in Tables 1, 2 in this paper are not consistent with those reported in other benchmarks like Pocket2Mol?
2. Could the authors provide more ablation experiments to demonstrate the effect of different parts of the FLAG on Table 2 results?
3. If limiting the size of molecule generation is dependent on the interaction with the target, would expanding the size of the pocket affect the quality of molecule generation?


**Summary Of The Paper:**

This paper tackles the problem of structure-based drug design. The authors propose a framework, called FLAG, where the molecules are generated fragment-by-fragment. The authors first reconstructed their motif vocabulary and then generate molecules while considering interactions with target atoms. The authors conduct various experiments on the benchmarks of structure-based molecular generation and achieved performance improvements on some metrics.

**Summary Of The Review:**

Overall, this work is borderline. The authors did a good job of combining motif-based and target-based approaches, and there are some performance improvements. However, the technical and theoretical contributions are insufficient. The ablation studies are not solid enough. Thus, I vote for a borderline rejection but may change the score during rebuttal.

---

> ### Author Response · Authors · 2022-11-12
> **Response to Reviewer xsvH**
>
> We thank the reviewer for the valuable suggestions and questions! As for the comments:
>
> Comment1: Could the authors explain why the results in Tables 1, and 2 in this paper are not consistent with those reported in other benchmarks like Pocket2Mol?
>
> Response1: We observe that the results in Tables1, and 2 are slightly different from the results reported in baseline papers. There are mainly two reasons: (1) As stated in Section 4.1, we additionally filter invalid molecules that can not be sanitized with RDkit in the dataset, which leads to a refined subset with around 160k data points (over 180k in Pocket2Mol). (2) There is non-negligible randomness in the process of molecule generation and evaluation. Following Pocket2Mol, we randomly sample 100 molecules for each protein pocket in the test set. Moreover, calculating metrics such as the vina score also introduces much variance.
> In our paper, we use the same training and test set for all the baseline methods. We use their open-source codes and follow their default settings. The results in Table 1,2 generally align well with results reported in other papers.
>
> Comment2: Could the authors provide more ablation experiments to demonstrate the effect of different parts of the FLAG on Table 2 results?
>
> Response2: Thanks for the constructive suggestion! Here we show the results of ablation studies regarding the structure refinement module (FLAG w/o SR) and the torsion angle module (FLAG w/o T). Specifically, in FLAG w/o SR, we remove the structure refinement in each generation step; in FLAG w/o T, we do not further rotate the motif for adjustment. Generally, we can observe that FLAG has the best performance compared with its variants, which again verifies the effectiveness of structure refinement and torsion angle module. We have also included the results in Table 2.
> | Angles | LiGAN | AR   | GraphBP | Pocket2Mol    | FLAG w/o SR | FLAG w/o T | FLAG          |
> |--------|-------|------|---------|---------------|-------------|------------|---------------|
> | CCC    | 7.26  | 2.17 | 1.67    | 0.88          | 0.30        | 0.25       | **0.24** |
> | CCO    | 8.31  | 1.95 | 2.02    | 0.93          | 0.17        | 0.19       | **0.12** |
> | CNC    | 6.90  | 2.36 | 1.26    | 0.55          | 0.24        | 0.22       | **0.18** |
> | OPO    | 5.81  | 2.83 | 1.75    | 0.32          | 0.34        | 0.31       | **0.25** |
> | NCC    | 7.53  | 3.59 | 2.10    | 0.68          | 0.29        | 0.26       | **0.21** |
> | CC=O   | 6.85  | 3.56 | 1.23    | 0.80          | 0.23        | 0.27       | **0.15** |
> | CCCC   | 1.26  | 0.81 | 0.79    | **0.73** | 0.80        | 0.82       | 0.77          |
> | cccc   | 6.43  | 8.26 | 5.41    | 4.30          | 0.33        | 0.31       | **0.30** |
> | CCCO   | 2.34  | 1.55 | 1.09    | 1.14          | 0.57        | 0.93       | **0.51** |
> | OCCO   | 1.62  | 1.79 | 1.47    | 1.73          | 1.39        | 1.72       | **1.27** |
> | Cccc   | 4.98  | 8.27 | 3.13    | 2.35          | 0.58        | 0.51       | **0.48** |
> | CC=CC  | 6.01  | 5.86 | 3.08    | 3.40          | 1.54        | 1.76       | **1.22** |
>
>
> Comment3: If limiting the size of molecule generation is dependent on the interaction with the target, would expanding the size of the pocket affect the quality of molecule generation?
>
> Response3: Thanks for the good question.
> Expanding the size of the pocket will incorporate more context information, but will introduce more computational costs as well. As implied by previous work [1], the size/volume of a binding pocket is dependent on the structure of the protein and is largely in a specific range. Moreover, previous works [2, 3] also indicate that the interactions between the molecule and target protein largely depend on the local regions of pockets. Therefore, we speculate that expanding the size of the pocket will have a limited influence on the quality of the generated molecules. We will conduct more analysis in our future work.
>
> [1]Liang et al., Anatomy of protein pockets and cavities: measurement of binding site geometry and implications for ligand design, Protein Science 1998.
>
> [2]Krivák et al., Improving protein-ligand binding site prediction accuracy by classification of inner pocket points using local features, Journal of Cheminformatics 2015
>
> [3]Luo et al., A 3d generative model for structure-based drug design. NeurIPS 2021
>
> Comment4: Reproducibility: N/A, the code has not been released.
>
> Response4: We have uploaded our code to https://anonymous.4open.science/r/FLAG-5C01/ for reference. It is our hope to benefit the drug design community. We are cleaning and organizing the code and will open-source the codes of FLAG with detailed documentation after acceptance.

---

> > ### Comment · Reviewer_xsvH · 2022-12-05
> > **Thanks for the reply**
> >
> > I thank the authors for the rebuttals and additional experiments in response to this and other reviews. The authors gave some explanations to my questions and addressed ablation issues. However, I remain concerned about the technical contributions and the comparison in Table 1. Note that all the baselines generated molecules with lower Vina scores than the test set, which is the opposite of the results reported in their original papers.
> >
> > Overall, I think the current manuscript studies an interesting problem. But the authors need more time to provide some in-depth comparisons to make it more solid. Therefore, I still tend to weak rejection.

---

> > > ### Author Response · Authors · 2022-12-05
> > > **Response to Reviewer xsvH**
> > >
> > > We thank the reviewers for their valuable comments!
> > >
> > > As for the concern of the vina scores in Table 1, we checked the baseline papers and found only Pocket2Mol reported lower average vina scores (higher affinities) than the test set. As for the other baseline methods, our results are consistent with their reported results (higher vina scores than the test set). In our work, we tried to reproduce the results of baseline methods with their open-source codes. Generally, the results are comparable to those reported in the original papers. As for the slight differences, we find the following factors that may influence the final results:
> > >
> > > (1) The dataset split is different. As stated in Section 4.1, we additionally filter invalid molecules that can not be sanitized with RDkit in the dataset, which leads to a refined subset with around 160k data points (over 180k in Pocket2Mol). We resample 100 molecules respectively as the validation and test set, which is also different from Pocket2Mol.
> > >
> > > (2) There is non-negligible randomness in the process of molecule generation. Moreover, calculating metrics such as the vina score introduces much variance. As indicated in Pocket2Mol and our work, the variance of the vina score is over 2.0. Therefore, it is understandable that there is some difference in vina scores.
> > >
> > > We hope our explanation could address the concern of reviewers. Besides the improvement of vina scores brought by our method FLAG, our work provides a proof-of-the-concept method to generate ligand molecules fragment-by-fragment. The generated molecules also exhibit higher validity and more realistic sub-structures. We will keep polishing our work to make it more solid!

---

### Official Review · Reviewer_fSis · 2022-10-24

**Confidence:** 2
**Correctness:** 4
**Technical Novelty And Significance:** 3
**Empirical Novelty And Significance:** Not applicable
**Recommendation:** 8

**Clarity, Quality, Novelty And Reproducibility:**

* The paper is clear to understand and the novel parts are not only significant but also effective.

**Strength And Weaknesses:**

[Strength]
* The overall framework is not only innovative but also effectively driven by the functional requirements of each step. For example, rotation angle prediction and structure refinement through force field are plausible and shown to be effective by the ablation study.
* The paper is easy to read and well organized.


[Weaknesses]
* 3hwf case in Figure 3: All generated molecules have lower SA scores than the reference molecule. Is there any chance that the proposed model will produce molecules that cannot be synthesized?
* Minor typo - Page 6. "we fix the the coordinates" -> "we fix the coordinates"


**Summary Of The Paper:**

This paper presents a new 3D molecule-generating model conditioned on a receptor. It generates a ligand molecule fragment-by-fragment, unlike other similar models that generate a molecule atom-by-atom. They first extract a set of motifs serving as building blocks of generation. Then context encoding, focal motif selection, next motif prediction and attachment, angle prediction, and refinement are followed. The experiment results show that it outperforms all the baselines in almost all metrics. Substructure analysis shows that it better captures the distributions of the dataset compared to other baselines.

**Summary Of The Review:**

Since the model is novel and effective, this paper will be a good addition to ICLR.

---

> ### Author Response · Authors · 2022-11-12
> **Response to Reviewer fSis**
>
> We thank the reviewer for the appreciation and valuable suggestions!
> As for the comments:
>
> Comment1: 3hwf case in Figure 3: All generated molecules have lower SA scores than the reference molecule. Is there any chance that the proposed model will produce molecules that cannot be synthesized?
>
> Response1: Thanks for the question! Since FLAG is not designed to explicitly optimize SA scores, the generated molecules may have low SA scores on some examples. However, we note that FLAG achieves the highest average SA score compared to baseline methods.
>
> Moreover, SA uses a template matching procedure to predict the synthetic accessibility from the frequency of the sub-structures. The statistics of sub-structures are calculated from 1 million molecules of PubChem in 2009 [1]. However, in today, the compounds in the Pubchem have reached over 112 million [2]. Therefore, lower SA scores do not necessarily indicate poorer synthetic accessibility, and the differences in synthetic accessibility between generated molecules and the reference molecule are not that large.
>
> In practice, we can optimize the generated molecules by experts or other optimization methods [3, 4] to improve synthetic accessibility. For example, as for the first molecule for the 3hwf case in Figure 3 (Oc1c(F)c(F)c2c(c1F)OC(F)O2), we can improve the SA score from 0.66 to 0.91 by simply pruning some side chains (Oc1ccc2c(c1)OCO2). Meanwhile, the other metrics are less influenced (QED: 0.58, Vina score: -6.3). We will explore molecule optimization modules for FLAG in future works.
>
> [1]Ertl et al., Estimation of synthetic accessibility score of drug-like molecules based on molecular complexity and fragment contributions, Journal of Cheminformatics 2009
>
> [2]https://pubchem.ncbi.nlm.nih.gov/
>
> [3] Chen et al., Molecule optimization by explainable evolution, ICLR 2021
>
> [4] Jin et al., Learning Multimodal Graph-to-Graph Translation for Molecule Optimization, ICLR 2019

---

> > ### Comment · Reviewer_fSis · 2022-12-13
> > **Thanks for the responses**
> >
> > Thanks for the detailed responses to all my questions. I have no concerns regarding the paper, Good luck!

---

### Author Response · Authors · 2022-11-29
**Thanks for your insightful comments and valuable suggestions**

Dear Reviewers,

Thanks again for your insightful comments and valuable suggestions, which are of great help to improve our work.  We sincerely hope that our rebuttal has properly addressed your concerns.  If not, please let us know your further concerns, and we will continue actively responding to your comments and improving our paper. We are looking forward to your further responses and comments.

Best,

Authors

---

### Decision · Program_Chairs · 2023-01-20

**Decision:**

Accept: poster

**Justification For Why Not Higher Score:**


Technical novelty is somewhat low since many of the steps and ideas are taken fairly directly with minor variations from previous papers. Similar approaches have been proposed before though comparison would be challenging due to lack of code availability.

**Justification For Why Not Lower Score:**


This is a well-engineered and executed solution that works well in comparison to (limited) baselines.

**Metareview: Summary, Strengths And Weaknesses:**


The goal is to create a 3D molecular binder to a given protein (pocket). The binder is created iteratively, one motif at a time. The key steps include protein and intermediate molecule encoding, focal motif selection, prediction of the new motif to add and how it is attached, torsion angle update, as well as an interleaving predicted-force guided 3D coordinate refinement step. The architecture is trained with masking and based on noise perturbed coordinates. The experiments are carried out relative to a subset of cross-docked dataset. Technical novelty is low since many of the steps and ideas are taken fairly directly with minor variations from previous papers though put together cleanly for the designed purpose here. Other related motif-fragment methods were available prior to this work but comparisons could be hampered by lack of code release. Overall, this is a well-engineered and executed solution that works well in comparison to (limited) baselines.


**Note From Pc:**

if the above contains the word "oral" or "spotlight" please see: "oral" presentation means -> notable-top-5% and "spotlight" means -> notable-top-25%. As stated in our emails, we are disassociating presentation type from AC recommendations